# Evaluating Evaluation Metrics: A Framework for Analyzing NLG Evaluation Metrics using Measurement Theory

**Ziang Xiao**[*,†,♠]**, Susu Zhang**[*,‡]**, Vivian Lai**[◇]**, Q. Vera Liao**[♠]

[†]Johns Hopkins University    [♠]Microsoft Research Montréal
[‡]University of Illinois Urbana-Champaign    [◇]Visa Research
ziang.xiao@jhu.edu    szhan105@illinois.edu
viv.lai@visa.com    veraliao@microsoft.com

## Abstract

We address a fundamental challenge in Natural Language Generation (NLG) model evaluation—the design and evaluation of evaluation metrics. Recognizing the limitations of existing automatic metrics and noises from how current human evaluation was conducted, we propose METRICEVAL, a framework informed by measurement theory, the foundation of educational test design, for conceptualizing and evaluating the *reliability* and *validity* of NLG evaluation metrics. The framework formalizes the source of measurement error and offers statistical tools for evaluating evaluation metrics based on empirical data. With our framework, one can quantify the uncertainty of the metrics to better interpret the result. To exemplify the use of our framework in practice, we analyzed a set of evaluation metrics for summarization and identified issues related to conflated validity structure in human-eval and reliability in LLM-based metrics. Through METRICEVAL [1], we aim to promote the design, evaluation, and interpretation of valid and reliable metrics to advance robust and effective NLG models.

## 1 Introduction

Evaluation metrics provide quantitative assessments to guide model development, benchmark scientific progress, and inform generalizability across tasks and domains (Novikova et al., 2017). Effective evaluation metrics can extract valuable signals and robust evidence from model outputs that describe model capability, diagnose model failures, and compare the strengths and weaknesses of different models, allowing for more informed decision-making in real-world deployment (Zhou et al., 2022). Conversely, problematic evaluation metrics can mislead model diagnoses, development, and deployment, resulting in downstream harms

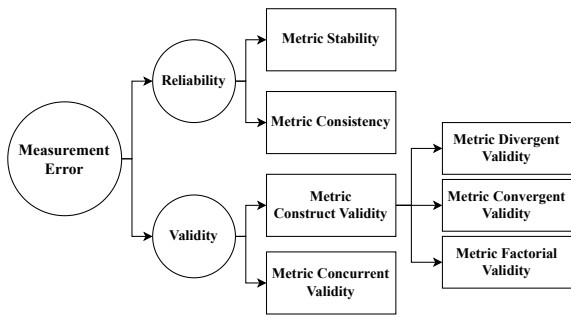

Figure 1: METRICEVAL: A framework that conceptualizes and operationalizes four main components of metric evaluation, in terms of *reliability* and *validity*

to individuals and society (Yeo and Chen, 2020; Sheng et al., 2021).

Designing effective evaluation metrics for natural language generation (NLG) tasks has long been challenging due to the complex nature of language, open-endedness of tasks, multifaceted and context-dependent definition of language quality (Nema and Khapra, 2018; Zhou et al., 2022; Gehrmann et al., 2022; Sai et al., 2022). Most recently, the NLG evaluation challenge has been further exacerbated by the emergence of "general-purpose" large language models (LLMs), further demanding evaluation methods to capture model utility for diverse downstream use cases. To address these challenges, researchers and practitioners have developed various types of NLG evaluation metrics, including word-based metrics (e.g., ROUGE, BLEU), embedding-based metrics (e.g., BERTScore, MoverScore) and to end-to-end metrics (e.g., BLEURT, G-Eval).

With this increasingly rich set of evaluation metrics being pursued, we must understand how *good* each metric is. While researchers pointed out shortcomings of popular metrics (e.g., ROUGE), such as their inability to capture semantic meanings, insensitivity to perturbations, and failure to reflect real-world performance (Sai et al., 2021; Liu

---

[*]Equal contribution.
[1]Data and code for analysis are available at: https://github.com/isle-dev/MetricEval

et al., 2016; Reiter, 2018; Celikyilmaz et al., 2020; Kauchak and Barzilay, 2006), there is a lack of principled approaches to evaluate NLG evaluation metrics, and to begin with, a lack of clear definition on what makes a metric good.

Some prior works have attempted to evaluate the quality of NLG evaluation metrics by their correlations with human judgments (Sai et al., 2021; Fabbri et al., 2021; Liu et al., 2016), which are deemed the gold standard for quality assessment. However, correlation with human preferences gives limited quality signals. More problematically, human evaluation data collection itself currently suffers from validation, standardization, consistency, and reproducibility issues (Clark et al., 2021; Howcroft et al., 2020; Belz et al., 2021; Khashabi et al., 2021). These issues subsequently undermine their validity as the foundation for evaluating automatic metrics.

In this paper, we introduce METRICEVAL, a theoretical framework to define the desiderata of and assess evaluation metrics by drawing from measurement theory in educational and psychological testing. Based on two core concepts in measurement theory that define a "good" metric in testing individual capabilities—*reliability* and *validity*, our framework conceptualizes and operationalizes four key desiderata: Metric Stability, Metric Consistency, Metric Construct Validity, and Metric Concurrent Validity. We further propose a set of statistical tools to quantify these desiderata to systematically evaluate evaluation metrics. METRICEVAL enables quantifying the standard error for each metric on a specific task which allows meaningful interpretations of the evaluation results. We demonstrate the utility of METRICEVAL with a case study of evaluating 16 metrics, including LLM-based metrics, on a summarization task.

This paper offers three contributions,

- Introduce and transfer metrics evaluation desiderata and methods from measurement theory in educational and psychological testing to NLG evaluation.

- Propose METRICEVAL, a theory-driven framework with a set of statistical tools for systematically analyzing and evaluating NLG metrics.

- A case study demonstrating how to apply our framework and identify issues of evaluation metrics for a summarization task.

## 2 Measurement Theory

Originating from educational and psychological testing, measurement theory aims to inform evaluation processes that devise a coherent numerical representation of individual capabilities—for instance, evaluating a person's language proficiency through essay responses to questions. Scores on these tests have direct consequences for high-stakes decisions, such as school admissions.

Key to measurement theory is the distinction between the *observed score* on a test, e.g., the score of an examinee's essay in a language proficiency exam, and the *true score* on the general *construct* (Cronbach and Meehl, 1955) that the test is theorized to measure, e.g., language proficiency. The gap between the observed and true scores is referred to as *measurement error* (Allen and Yen, 2001). Measurement theory defines two sources of measurement error, random and systematic. Random measurement errors are fluctuations specific to a time, place, examinee, and exam question that are transient and balance out to $0$ over repeated measures, and they have direct consequences on the *reliability* of the evaluation process. Systematic errors, on the other hand, are persistent shifts across one or more time, place, examinee, or exam questions, and they have direct consequences on test *validity* by producing observed scores with systematic deviations from the true score that the test purports to measure (e.g., a downward bias if the rubric on a language proficiency exam looks for specialized knowledge about a certain subject).

By evaluating and identifying the source of measurement errors, the test designer could iteratively improve their test design by adding or removing test items or changing their rubric. The results can also help the evaluator to interpret a test score with caution. For example, we can quantify the uncertainty as the standard error for meaningful comparison: when the difference between two models' metric scores on a benchmark does not afford conclusions about significant differences, the evaluator may consider narrowing the confidence interval by averaging scores from repeated measurements.

In short, as a safeguard to the trustworthiness of tests, measurement theory offers a conceptual framework for how the validity and reliability of a test should be formalized, evaluated, and optimized to reduce measurement error with the aid of statistical methods and tools.

## 2.1 Transferring Measurement Theory to the Context of NLG

There are obvious analogies between the measurement of human capability and the evaluation of NLG models. First, NLG evaluation is often performed via benchmarking. A benchmark data set is similar to an educational test consisting of a tailored collection of test examples, where question-specific scores are calculated based on predefined evaluation metrics (similar to scoring rubrics for human testing) and are aggregated into an overall score for the model. Second, when evaluating a model, we similarly hope to derive scores based on a candidate model's observed performance on a benchmark, so as to (1) draw inferences about unobservable capability in a specific domain (e.g., summarization) and (2) provide guidance on the model's expected behavior in future tasks of the domain. Similar to educational testing, the score is often interpreted and used beyond its nominal meaning, i.e., implying the model's general performance beyond the particular benchmark dataset. As more models are considered "general-purpose" models, there is an increasing need for measuring an NLG model's unobservable capability.

The conceptual and statistical tools provided by measurement theory can be transferred to assist in evaluating NLG metrics, specifically to quantify and identify different sources of measurement errors. Not only can these tools help the community systematically assess the shortcomings of evaluation metrics and identify misleading ones, but they also guide the interpretation of their evaluation results, as well as the re-design of existing metrics and the development of new ones. In the next section, we elaborate on how we transfer these tools from measurement theory to a framework that defines and assesses the reliability and validity of NLG evaluation metrics, and how they may help us interpret and improve NLG evaluation.

## 3 Metric Evaluation Framework

In this section, we introduce METRICEVAL and its components that define and assess different aspects of the "goodness" of NLG evaluation metrics, inspired by the core concept of reliability and validity in measurement theory (See Fig.1). METRICEVAL aims to evaluate and compare the reliability and validity of the metrics. For example, to evaluate a summarization model, one can apply reference-based metrics (e.g., ROUGE, BertScore), reference-free metrics (e.g., SUPERT), or human ratings on specific output quality aspects (e.g., coherence or relevance) on the same benchmark to draw inferences about model capability. For the remainder of this section, we will illustrate our framework with this running example of evaluating summarization models with diverse metrics.

It is important to note that the quality of evaluation results is also dependent on the *chosen* dataset and reference (for reference-based metrics), which, in NLG evaluation, are concerned with benchmark designs. Measurement errors may cascade from those components to the observed score. In this work, we focus on the metric aspect and answer questions such as "giving a CNN/Daily Mail benchmark, whether using ROUGE or BertScore or human ratings offer reliable and valid evaluation results.". This is an important question given the far-reaching impact that prevalent benchmarks can have on the output of the research community.

## 3.1 Reliability

The reliability of a metric is the extent to which the result is subject to *random* measurement error and thus *(in)consistent* across repeated measures, such as different (sub-)datasets within a benchmark or different raters scoring the model's output in human evaluation. Suppose two NLG models are scored on their performance based on Metric-A on a summarization benchmark. Researchers and practitioners often use the scores to draw inferences about the models' (relative) performances. When the two models are reported to differ in their scores (e.g., Metric-A = .39 vs. .42), a natural question is how much this reflects actual differences (true signal) versus fluctuations due to random measurement error (noise). If Metric-A is unreliable, the measurement error may mislead the comparison.

Sources of random measurement error that impair the reliability of a metric may include:

- Non-deterministic algorithms of some metrics may produce score variations on the same model outputs.

- The subsets of data points (e.g., different genres of articles) included in the benchmark.

- For human evaluations, the variability across raters, resulting from their subjectivity, inconsistency, errors, and so on.

In classical test theory (Spearman, 1904), the observed metric score of a model ($X$) is equal to

the sum of true score $T$ and error ($E$), which is assumed to be independent of $T$ and fluctuates around 0 with variance $\sigma_E^2$. The goal of evaluating a metric's reliability is hence to quantify the expected amount of uncertainty in the observed score due to random measurement error, known as the standard error of measurement ($\sigma_E$).

Empirical estimation of $\sigma_E$ is done via the reliability coefficient of a metric, denoted $\rho_{XT}^2 \in [0, 1]$. Formally, the reliability coefficient is defined as the proportion of variance in the observed score explained by the variance in the true score across NLG models rather than error, or equivalently, the squared correlation between $X$ and $T$:

$$\rho_{XT}^2 = \frac{\sigma_T^2}{\sigma_X^2} = 1 - \frac{\sigma_E^2}{\sigma_X^2}. \tag{1}$$

Metrics with higher reliability coefficients are more desirable. However, in reality, neither $T$ nor $E$ is observed. The reliability coefficient in Equ. 1 cannot be directly computed and is statistically approximated via several possible estimators.

METRICEVAL proposes to estimate the reliability coefficient from both *Metric Stability* and *Metric Consistency*. They reflect different reliability issues that can arise in different types of metrics, as we elaborate below. By quantifying and identifying reliability issues, metric developers can improve the scoring algorithms, and metric users can make more informed decisions in choosing metrics, interpret performance differences, and adopt mitigation strategies, e.g., increasing the test set size to mitigate consistency issues (Spearman, 1910).

### 3.1.1 Metric Stability

Metric Stability refers to how a metric score may fluctuate when evaluated again on the same model output. While we would expect perfect stability (i.e., $\sigma_E = 0$) for deterministic metrics, such as ROUGE-1 (Lin, 2004), the stochastic nature of some metrics (e.g., G-Eval (Liu et al., 2023)) may produce undesirable fluctuations when evaluating the same model outputs. As we see increasing use of automatic evaluation metrics with built-in stochasticity (e.g., LLM-based metrics), the stability of an evaluation metric in producing consistent scores on an output from one replication to another will be increasingly relevant.

We propose to quantify metric stability via the test-retest reliability coefficient: on the output generated by $N$ models, we compute the metric score with the same output twice for each model. Across different models, the Pearson correlation between the two sets of scores is the test-retest reliability coefficient. One can show that this correlation is an estimate of the reliability coefficient $\rho_{XT}^2$ as defined in Equ. 1. This is because, for the two metrics scores for a model, $X_1 = T_1 + E_1$ and $X_2 = T_1 + E_2$, the correlation in the observed scores, $\rho_{X_1 X_2}$, is algebraically equivalent to $\rho_{X_1 T_1}^2$, under the assumption that each model's true score doesn't change (i.e., $T_1 = T_2$) and that the expected fluctuation in metric evaluation remains the same (i.e., $\sigma_{E_1} = \sigma_{E_2}$) across the two evaluations (see derivations in Allen and Yen, 2001).

### 3.1.2 Metric Consistency

Metric Consistency describes how the metric score fluctuates within a benchmark dataset, i.e., across data points. If the metric score computed on each individual data point (e.g., summarization of a specific news article) deviates substantially from the average score across the benchmark dataset (e.g., across 100 news articles), the metric score is less reliable, in that it is more sensitive to perturbations in the specific data points employed in the benchmark dataset. In this case, for a specific model, the average metric score on any subset of tasks (e.g., 50 out of 100 news articles) is expected to be sensitive to the choice of included examples, and a good proportion of difference across two evaluated models' average scores would also be attributed to this noise. Drawn from the estimation of internal consistency reliability in measurement theory, the estimation of metric consistency depends on the degree to which scores from different subsets of the benchmark dataset agree with one another.

The coefficient $\alpha$ (Cronbach, 1951) provides a measure of Metric Consistency. Let $J$ denote the total number of data points in the test dataset, $Y_j$ the observed score (of a model) on the $j$th data point alone (e.g., the $j$th news article), and $X = \sum_{j=1}^J Y_j$ the overall score of the model on the full test set. Then $\alpha$ provides a lower bound to the true reliability of $X$, i.e.,

$$\rho_{XT}^2 \geq \alpha = \frac{J}{J-1}\left[\frac{\sigma_X^2 - \sum_{j=1}^J \sigma_{Y_j}^2}{\sigma_X^2}\right], \tag{2}$$

where $\sigma_{Y_j}^2$ is the variance of $Y_j$ across models. Equality holds when all the individual data point scores ($Y_j$s) have equal correlations with the true score ($T$), which may be violated in practice, leading to the underestimation of true reliability via the

coefficient $\alpha$ formula.

## 3.2 Validity

Validity is another core component of METRICE-VAL. Metrics with low validity lead to *systematic* measurement errors that deviate the observed score from the true score that the test purports to measure. In other words, benchmarking is valid only when the metric scores can inform their intended interpretations (e.g., model capability) and uses (e.g., predicting models' real-world behavior).

Our framework is theoretically grounded in Messick's unified theory of test validity (e.g., Messick, 1995), under which the emphasis is given to the validation of *inferences drawn* from the test score, rather than the validation of the test itself. Different types of validities should be recognized as possible ways to gather supporting evidence for intended inferences (interpretations and uses) from the metric score. Our framework conceptualizes two types of a metric's validity, concurrent validity, and construct validity (e.g., Allen and Yen, 2001), which can be applied in different situations—when a validated reference criterion is available or not—as we elaborate below.

### 3.2.1 Metric Concurrent Validity

Metric Concurrent Validity relies on another validated metric as the reference criterion. This type of validity is most relevant when evaluating a metric as an alternative to existing ones that may be expensive and infeasible to acquire in practice. For example, evaluations by trained human experts are often challenging at a large scale, motivating the development of automatic alternatives. One can conclude that an automatic metric is a valid proxy if it has high concurrent validity using the expert valuation results as the reference criterion.

When both the target evaluation metric ($X$, e.g., a new automatic metric) and the reference criterion ($Y$, e.g., expert evaluation) are continuous, a straightforward way to quantify concurrent validity is via their Pearson correlation, $\rho_{XY}$, often referred to as the (criterion-related) validity coefficient. One should note that measurement error in either $X$ or $Y$ is expected to attenuate this correlation (Spearman, 1910): At the population level, $\rho_{XY}$ is bounded above by the square root of the product of the two scores' ($X$ and $Y$) *reliabilities*. This again highlights the importance of safeguarding the reliability of the evaluation metric, as a noisy metric with low reliability is expected to yield poor predictive power on the criterion of interest.

### 3.2.2 Metric Construct Validity

Construct validity, a term coined by Cronbach and Meehl (1955), refers to the degree to which the observed behaviors on the test (e.g., test scores) can reasonably reflect the intended construct (e.g., language proficiency). This notion is directly applicable to evaluation metrics that are *explicitly* constructed to assess specific aspects of a model's performance or output quality, e.g., human evaluation (or automatic metrics, if developed specifically) on summarization, coherence, fluency, consistency, etc. However, even for metrics of which the intended construct is not explicitly defined, it is still necessary to understand what underlying dimensions of model capabilities they *actually* capture.

It is important to note that the underlying construct is often latent and not directly observable to assess its relation with the measure. Measurement theory, therefore, provides statistical tools to assess the construct validity of a measure through its relation with other observable variables (e.g., other tests purported to reflect the same or different constructs). We consider three such aspects of validity based on the measurement literature:

- *Metric Convergent Validity*: Whether metrics of identical or related construct(s) are indeed related. For example, for the same aspect of summarization quality (e.g., coherence), scores provided by different evaluation methods (e.g., by different raters) should be highly correlated.

- *Metric Divergent Validity*: Whether metrics of unrelated constructs are indeed unrelated. For example, for distinct aspects of summarization quality (e.g., coherence and relevance), scores provided by the same method (e.g., by the same rater) should show substantially lower correlations than those for the same quality across methods. Low divergent validity could indicate method bias: e.g., the observed score depends greatly on the rater's subjective tendency rather than the model's performance on the rated dimension.

- *Metric Factorial Validity*: Whether the observed metric scores align with the theory about unobserved factors underlying the scores. For example, if scores on multiple evaluation metrics exhibit high correlations,

this might suggest the presence of a common underlying factor causing these scores to move in unison.

We introduce two statistical tools to evaluate these aspects of construct validity. Specifically, Metric Convergent Validity and Metric Divergent Validity can be evaluated through the analysis of a multitrait-multimethod (MTMM) table, and Metric Factorial Validity can be evaluated via factor analysis. Note that these validity evaluation methods will only inform if there is an underlying construct or how many of them are being captured. Defining *what* these constructs are will require further conceptualization and theorizing.

**The MTMM table**   presents a way to scrutinize whether observed metric scores act in concert with theory on what they intend to measure, when two or more constructs are measured using two or more methods (Campbell and Fiske, 1959). For example, when evaluating a summarization model, researchers may ask several raters to rate the generated outputs on four "traits" (aspects of output quality), e.g., coherence, consistency, fluency, and relevance. In this case, the MTMM table allows examining whether, across different raters (evaluation methods), the raters' scores indeed appear to characterize the model's performance on four distinct constructs. By convention, an MTMM table reports the pairwise correlations of the observed metric scores across raters and traits on the off-diagonals and the reliability coefficients of each score on the diagonals. The analysis of an MTMM table is exemplified in Sec. 4.1.2.

**Factor Analysis**   examines *Metric Factorial Validity* (e.g., Thurstone, 1947) when the observed metric scores are assumed to measure a smaller number of unobserved factors. For example, if scores from multiple evaluation metrics exhibit high correlations, this might suggest the presence of a common underlying factor causing these scores to move in unison. Under a factor analysis model, the distribution of the observed score on an indicator $X_j$, such as a particular evaluation metric, is a function of a linear combination of the model's factor scores on $K \geq 1$ general latent factors $(f_1, \ldots, f_K)$ and the unique score $U_j$ on the indicator $j$ unexplained by the latent factor, including measurement error, i.e.,

$$X_j = f(\lambda_{j1}f_1 + \ldots + \lambda_{jK}f_K + U_j). \quad (3)$$

$f(\cdot)$ can be the identity function for normally distributed observed scores, but when scores are ordinal (e.g., expert ratings on a 5-point scale) or skewed, we suggest adopting an ordinal factor model (Muthén, 1984) where $f(\cdot)$ is a step function that evaluates whether Equ. 3 exceeds specific thresholds for each score category on the latent continuum. Factor analysis can be exploratory or confirmatory. In the latter, select loadings ($\lambda_{jk}$s) are constrained to 0 to represent the theorized nomological network, e.g., an expert rating on consistency loads on no other dimensions. By establishing the *Metric Factorial Validity* through factor analysis, we could further develop more effective metrics by answering the following questions:

- Fit indices: For confirmatory factor analysis, how well does the theorized factorial structure align with the observed data?

- Factor scores: What is an NLG model's factor score on a particular dimension?

- Factor loadings: How much does a specific factor affect an observed metric score?

- Residual correlation: For different evaluation metrics, are the residuals (unexplained score variation by the common factors) correlated, which may suggest additional dimensions?

## 4   Case Study

To illustrate how to apply METRICEVAL to evaluate NLG evaluation metrics, in this section, we ran a case study on evaluating summarization metrics. As noted earlier, our evaluation focuses on the metrics and the results should be interpreted as dependent on the benchmark dataset used. We leave it for future research to explore the generalizability of the results across different benchmark datasets.

### 4.1   Summarization Metric Evaluation

We analyzed the SummEval dataset (Fabbri et al., 2021), a benchmark for summarization tasks. The benchmark contains 1700 summaries generated by 17 models on the CNN/Daily Mail dataset. In this dataset, each generated summary was rated by three experts, who provided 5-point-scale ratings on four dimensions: Coherence, Consistency, Fluency, and Relevance. We ran 16 types of popular automatic metrics that include rule-based metrics, embedding-based metrics, end-to-end metrics, and LLM-based metrics that are reference-based or reference-free,

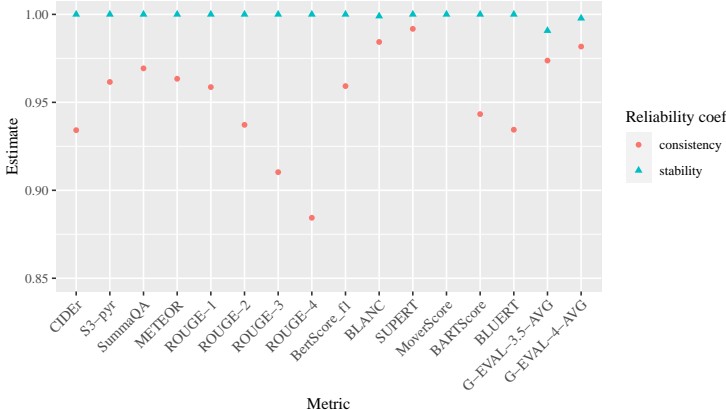

Figure 2: Estimated Metric Stability and Metric Consistency of common summarization metrics.

see Appx.A.1. Since the score distributions on many evaluation metrics were skewed, we normalized automatic evaluation scores for the subsequent analyses, see Appx.A.2.

### 4.1.1 Metric Stability and Consistency

To evaluate an automatic metric's stability, we computed the metric score twice for each model's output on each data point, calculated two sets of average scores for each model, and reported the correlation between the two sets of scores on the 17 models. A metric's consistency was evaluated via the coefficient $\alpha$ in Equ. 2. Fig. 2 presents the Metric Stability and Metric Consistency estimates of select metrics (full results in Fig. 5 in Appx). Most metrics achieved high stability. Metrics with non-deterministic algorithms, such as LLM-based metrics G-Eval, displayed higher levels of measurement error in terms of Metric Stability. While compared to G-Eval with GPT3.5, G-Eval with GPT-4 yields higher stability. For Metric Consistency, for the ROUGE family, a longer n-gram makes the metric less reliable and more prone to potential data perturbations in the test dataset. Therefore, to mitigate measurement error, for less stable LLM-based metrics, the metric user should consider aggregating scores over multiple runs, and for less consistent metrics such as ROUGE-4, the evaluator should consider using a larger test dataset. To illustrate, Fig. 3 shows the relationship between G-Eval metric consistency and test dataset size.

Conventionally, a reliability coefficient above .9 indicates good reliability. The metric stability and consistency estimates can help approximate the standard error of measurement of an average test dataset metric score $(X)$, by observing from Equ. 1 that $\sigma_E = \sigma_X \sqrt{1 - \rho_{XT}^2}$. For example, the sam-

ple standard deviation in the average test dataset METEOR score was .38 and the metric consistency estimate was .966, translating to an expected measurement error due to score variability across the 100 data points of $.38 \times \sqrt{1 - .966} \approx .07$. A METEOR score difference between two models less than .07 would thus be of limited interest, as the difference is less than the expected amount of fluctuation in the score due to measurement error.

### 4.1.2 Metric Construct Validity

We begin by evaluating the construct validity of expert ratings in the SummEval dataset. These evaluations were conducted in a confirmatory manner, assuming that the four ratings provided by each expert on a summarization output's Coherence, Consistency, Fluency, and Relevance indeed measure the four distinct dimensions. Tab. 1 presents the MTMM table for the three expert's ratings on four dimensions. Metric Convergent Validity can be examined by inspecting the **bolded** entries: Inter-rater agreements based on Kendall's $\tau$ on the same dimension were high $(.40 - .81)$ in general but lower for Fluency $(.40 - .63)$. *Italic* entries can inform the evaluation of metric divergent validity: Overall, an expert's ratings on different dimensions showed lower correlations than ratings by different experts on the same dimension, with the exception of Coherence and Relevance, which sometimes showed correlations (underscored, $.65 - .71$) nearly as high as those on ratings for Coherence (or Relevance) across raters $(.69 - .81)$. This may suggest that, although the expert raters were asked to separately rate Coherence and Relevance, they might inherently be rating the summarization outputs on the same underlying characteristic.

Confirmatory factor analysis was further con-

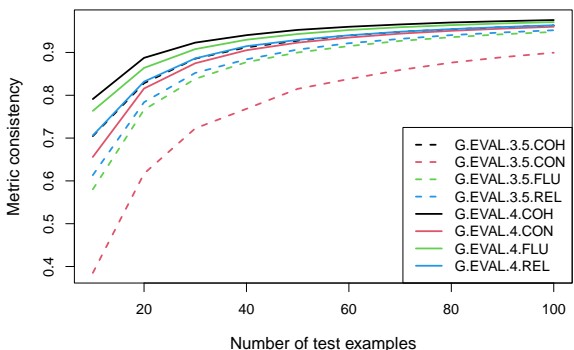

Figure 3: Estimated Metric Consistency for G-EVAL Metrics by number of data points in evaluation benchmark.

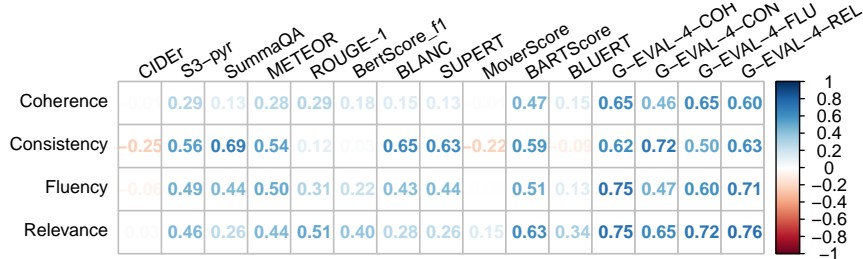

Figure 4: Concurrent validity coefficients of the selected metrics in predicting the four expert-rated dimensions' factor scores. Values are based on Kendall's $\tau$.

ducted (see Appx. A.4) to test the observed conflated validity structure indicated by the MTMM analysis. The results show that the four-factor model fitted the observed data adequately well (Comparative Fit Index = .999, Tucker-Lewis Index = .999, Root Mean Square Error of Approximation = .047 < .05), supporting the theorized loading structure, i.e., experts indeed rated on four factors. However, the estimated factor correlations suggested high correlations between dimensions, especially for Coherence and Relevance. This result supports a conflated validity structure.

Since Coherence and Relevance are distinct by definition (Fabbri et al., 2021), the conflated validity structure indicates potential issues in the expert rating process. Such issues may cascade if new automatic evaluation metrics were trained or validated on these expert ratings. In this case, several remedial steps are advised, including (1) revisiting the dimensions' conceptual distinctiveness and, if needed, revising the theoretical framework; (2) the human-annotation guidelines could be reviewed; and (3) the test set could be examined to assess its ability to distinguish model performance across dimensions. If these actions are insufficient, we recommend that the community consider alterna-

tive conceptualizations of summarization quality, as suggested in recent works (Liu et al., 2022; Clark et al., 2023).

## 4.2 Metric Concurrent Validity

For each automatic evaluation metric, we evaluated its concurrent validity: i.e., should the metric score be used to predict expert rating on a dimension as a more cost-efficient alternative? Different from prior studies (Fabbri et al., 2021), we report Kendall's $\tau$ between each model's metric scores and the factor scores (instead of the raw means) based on expert ratings on the four dimensions. The metric concurrent validity coefficients are presented in Fig. 4 (see full results in Appx Fig. 8).

Although BARTScore and G-Eval were sensitive to detecting quality signals in all four expert-rated dimensions, the lack of variance in the validity coefficients surfaces another issue—their lack of capability to distinguish different dimensions. For example, although G-EVAL-4-COH is designed for Coherence, it strongly correlated with Fluency (.75) and Relevance (.75). As a reference, its correlation with Coherence was .65, and the correlation between G-EVAL-4-FLU (designed to assess Fluency) and Fluency was only .60. This in-

| | | COH | | | CON | | | FLU | | | REL | | |
|---|---|---|---|---|---|---|---|---|---|---|---|---|---|---|
| | | Expert 1 | Expert 2 | Expert 3 | Expert 1 | Expert 2 | Expert 3 | Expert 1 | Expert 2 | Expert 3 | Expert 1 | Expert 2 | Expert 3 |
| COH | Expert 1 | 0.96 | **0.79** | **0.69** | *0.35* | 0.57 | 0.28 | *0.59* | 0.26 | 0.54 | *0.71* | 0.56 | 0.72 |
| | Expert 2 | - | 0.98 | **0.81** | 0.17 | *0.41* | 0.19 | 0.56 | *0.21* | 0.51 | 0.74 | *0.65* | 0.66 |
| | Expert 3 | - | - | 0.95 | 0.18 | 0.42 | *0.09* | 0.6 | 0.34 | *0.38* | 0.6 | 0.6 | *0.65* |
| CON | Expert 1 | - | - | - | 0.98 | **0.74** | **0.79** | *0.44* | 0.48 | 0.46 | *0.36* | 0.41 | 0.33 |
| | Expert 2 | - | - | - | - | 0.98 | **0.68** | 0.64 | *0.44* | 0.6 | 0.57 | *0.67* | 0.55 |
| | Expert 3 | - | - | - | - | - | 0.98 | 0.36 | 0.34 | *0.52* | 0.39 | 0.43 | *0.3* |
| FLU | Expert 1 | - | - | - | - | - | - | 0.97 | **0.53** | **0.63** | *0.65* | 0.74 | 0.72 |
| | Expert 2 | - | - | - | - | - | - | - | 0.96 | **0.4** | 0.38 | *0.41* | 0.37 |
| | Expert 3 | - | - | - | - | - | - | - | - | 0.95 | 0.63 | 0.57 | *0.59* |
| REL | Expert 1 | - | - | - | - | - | - | - | - | - | 0.92 | **0.79** | **0.78** |
| | Expert 2 | - | - | - | - | - | - | - | - | - | - | 0.98 | **0.72** |
| | Expert 3 | - | - | - | - | - | - | - | - | - | - | - | 0.91 |

Table 1: Multitrait-Multimethod table of expert ratings.
*Notes:* Diagonal entries are metric consistency coefficients between 0 and 1. Entries in **bold** are the correlations of ratings on the same dimension by different experts. Entries in *italic* are the correlations on different dimensions by the same expert. Underscored entries coherence and relevance rating correlations by the same expert, which showed strong correlations.

dicates a systematic discrepancy between the G-EVAL-4-COH score and what it purports to measure — Coherence instead of Fluency/Relevance. The MTMM table corroborates this lack of divergent validity for expert-based and G-EVAL metrics (Appx. Tab. 2). Correlations in G-EVAL scores across dimensions frequently exceeded .70 (underscored *italic* entries), often exceeding the correlations on the same rated dimension across methods (**bolded**). On the contrary, SummaQA only reacts to Consistency which makes it a more desirable metric for Consistency even though its correlation with expert rating was slightly lower than G-EVAL-4-CON. This might guide the refinement/disambiguation of prompts for LLM-based evaluations, in search of one that correlates strongly with the target dimension but is less confounded by the other nuisance.

### 4.3 Summary of the Case Study

Our findings indicate that metrics based on LLMs exhibited lower stability, some below the conventional threshold of .9. In the ROUGE metric family, an increase in $n$-gram length was associated with decreased metric consistency, heightening susceptibility to benchmark task perturbations. Both MTMM and factor analysis identified a conflation between expert ratings of Coherence and Relevance. Lastly, while BARTScore and G-Eval demonstrated high agreements with expert-rated dimensions, the lack of variability in metric concurrent validity suggested a lack of differentiation between theorized dimensions.

## 5 Related Work

NLG evaluation metrics undergo validation through various methods. The most widely used method is examining correlation with human judg-

ments (Sai et al., 2021; Fabbri et al., 2021; Liu et al., 2016). Beyond correlation, Ni'mah et al. (2023) proposed a comprehensive framework checklist, aiming to verify the faithfulness of automatic metrics to human preferences at both aspect and system levels. Fomicheva and Specia (2019) analyzed the local dependency between metric and human judgments and looked into the consistency of human evaluation. However, the inconsistency and subjectivity of human judgment, in addition to the non-transparent and non-standardized annotation process (Sai et al., 2021; Liu et al., 2016; Reiter, 2018; Celikyilmaz et al., 2020; Kauchak and Barzilay, 2006; Sai et al., 2022), create a shaking foundation. Another approach to evaluating metrics is data perturbation and resampling (Caglayan et al., 2020; Sai et al., 2021; Deutsch et al., 2021). Such a method can diagnose a metric's consistency and robustness across out-of-distribution datasets. In addition, researchers conducted qualitative analysis (Zhang et al., 2019; Tao et al., 2018; Hanna and Bojar, 2021). Although qualitative analyses provide in-depth insights, they are not scalable and cost-efficient. Closely aligned with our efforts, Von Däniken et al. (2022) introduced a theoretical framework to examine the reliability of binary metrics.

## 6 Conclusion

Evaluation metrics inform model capability and guide model development. Drawing from the core concept of *reliability* and *validity* in measurement theory, we present METRICEVAL, a framework that conceptualizes and operationalizes four key desiderata for NLG metrics. With a collection of statistical tools, METRICEVAL offers the community an effective and principled way to analyze, evaluate and understand NLG evaluation metrics.

## Limitations

Evaluating evaluation metrics for NLG models should not be treated as a single-shot task. Instead, as suggested in Messick's unified theory of validity (Messick, 1995), it is essential to continuously gather cumulative evidence of validity to ensure the ongoing effectiveness and reliability of the metrics. The process of accumulating valid evidence is an iterative and dynamic endeavor that aligns with the evolving landscape of NLG models and their applications. Future studies are necessary to collect other types of evidence, such as a metric's ability to predict users' preferences, to continuously evaluate the effectiveness of an NLG metric.

Measurement errors may surface and accumulate at every stage of the evaluation process, including benchmark design, data collection, etc. To perform the analysis of evaluation metrics, we have to assume the reliability and validity of the other parts of the evaluation process. Therefore, the results of the case study should be interpreted as dependent on the benchmark used, e.g., CNN/Daily Mail dataset. Future study is required to study the generalizability of the results across different benchmarks.

Our framework does not aim to provide comprehensive coverage of all measurement error sources in NLG evaluation metrics. For example, we did not discuss predictive validity in our framework despite its importance in education and psychological testing. We encourage researchers and practitioners to extend our framework for other types of reliability and validity and build datasets to support more comprehensive analysis, e.g., a dataset with the model's real-world performance, to deepen our knowledge of NLG metric evaluation.

## Acknowledgements

We would like to thank the reviewers for their thought-provoking comments as well as Nicolas Le Roux and Jackie Chi Kit Cheung for their helpful discussions and feedback.

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

# A Appendix

## A.1 Metrics

Our selection of evaluation methods includes popular metrics for NLG tasks including both reference-based and reference-free metrics. Compared to the original SummEval dataset, we additionally selected end-to-end metrics and recent LLM-based metrics.

**ROUGE (Lin, 2004)** evaluates the generated summary by comparing the number of overlapping word sequences (n-grams) with a set of reference summaries.

**ROUGE-WE (Ng and Abrecht, 2015)** expands on ROUGE by incorporating soft lexical matching, which utilizes the cosine similarity of Word2Vec (Mikolov et al., 2013) embeddings.

**S3 (Peyrard et al., 2017)** is a model-based metric that combines existing evaluation metrics like ROUGE, JS-divergence, and ROUGE-WE. It utilizes these metrics as input features to predict the evaluation score.

**BertScore (Zhang et al., 2019)** calculates similarity scores by aligning the generated and reference summaries at the token-level. Token alignments are determined greedily to maximize the cosine similarity between contextualized token embeddings from BERT (Devlin et al., 2018).

**MoverScore (Zhao et al., 2019)** quantifies the semantic distance between a summary and a reference text by utilizing the Word Mover's Distance (Kusner et al., 2015). This distance measure operates over n-gram embeddings obtained from BERT representations.

**SummaQA (Scialom et al., 2019)** utilizes a BERT-based question-answering model to respond to cloze-style questions using generated summaries. This metric provides both the F1 overlap score and the confidence of the QA model.

**BLANC (Vasilyev et al., 2020)** is a reference-less metric that assesses the performance improvement of a pre-trained language model when provided with a document summary while performing language understanding tasks on the original document's text.

**SUPERT (Gao et al., 2020)** is a reference-less metric that measures the semantic similarity between model outputs and pseudo-reference summaries generated by extracting significant sentences from the source documents using soft token alignment techniques.

**BLEU (Papineni et al., 2002)** is a metric that focuses on precision at the corpus level. It calculates the n-gram overlap between a candidate utterance and a reference utterance while incorporating a penalty for brevity.

**CHRF (Popović, 2017)** measures character-based n-gram overlap between model outputs and reference documents.

**METEOR (Banerjee and Lavie, 2005)** determines an alignment between candidate and reference sentences by mapping unigrams in the generated summary to 0 or 1 unigrams in the reference, taking into account stemming, synonyms, and paraphrases.

**CIDer (Vedantam et al., 2015)** calculates the co-occurrence of 1-4 gram units between the candidate and reference texts, giving less weight to common n-grams and computing cosine similarity between the n-grams of the candidate and reference texts.

**BARTScore (Yuan et al., 2021)** evaluates text directly based on the probability of being generated from or generating other outputs. It addresses the modeling challenge using a pre-trained sequence-to-sequence (seq2seq) model called BART (Lewis et al., 2019).

**BLEURT (Sellam et al., 2020)** is a BERT-based metric that can model human judgments with a few thousand training examples, which may introduce some bias.

**G-Eval (Liu et al., 2023)** is a framework that leverages LLM with Chain-of-Thoughts (CoT) (Wei et al., 2022) to evaluate the quality of generated text. The generated outputs are assessed using a set of prompts along with generated CoT.

**Data Statistics (Grusky et al., 2018)** define three measures of dataset extractiveness: extractive fragment coverage, density, and compression ratio. Extractive fragment coverage quantifies the percentage of words in the summary that are derived from the source article, indicating the degree to which the summary is a derivative of the original text. Density represents the average length of the extractive fragment to which each summary word belongs. Compression ratio measures the word ratio between the articles and their summaries.

## A.2 Metric Normalization

Initial exploratory analysis revealed that the score distributions on many evaluation metrics were skewed. We thus normalized each automatic evaluation score (via the transformation $X_j^* = \Phi^{-1}(\frac{1}{N}\sum_{i=1}^{N}\mathcal{I}(X_{ij} \leq X_j))$) and subsequently worked with normalized automatic metric scores, which approximately followed $N(0,1)$ distribution and are more appropriate for correlational analysis and linear models.

## A.3 Metric Stability and Consistency Results

Fig. 5 presents the Metric Stability and Metric Consistency estimates of all automatic evaluations and the metric consistency estimates of all expert and automatic metrics.

## A.4 Confirmatory Factor Analysis on Expert Ratings

Confirmatory factor analysis was further conducted on the 12 (3x4) expert ratings, assuming that each rating loads only on the corresponding dimension. Given that the expert ratings were highly skewed (see Fig. 6 in Appx.), an ordinal factor model (Muthén, 1984) was fitted. Judging from commonly used fit indices (Comparative Fit Index = .999, Tucker-Lewis Index = .999, Root Mean Square Error of Approximation = .047 < .05), the four-factor model fitted the observed data adequately well, supporting the theorized loading structure, e.g., Experts rated on four factors. Tab 3 in Appx reports the estimated factor loadings and thresholds of each expert rating, assuming that

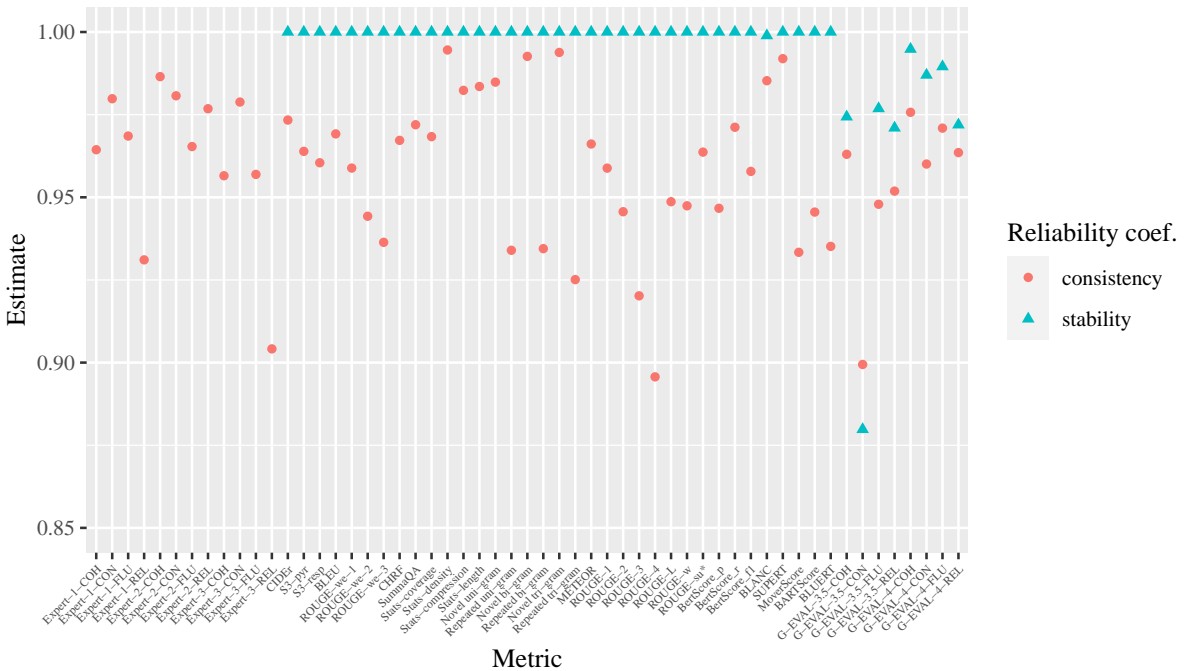

Figure 5: Metric stability and consistency estimates for all expert- and metric-based scores.

the four latent factors each have mean 0 and SD of 1. Loadings were generally high (adopting the $\geq .4$ convention) but varied across experts and dimensions (generally lower for Relevance). Rater differences were also found in their leniency: For instance, expert 3 was more likely than experts 1 and 2 to provide a rating of 5 (with lower threshold estimates for score 5) on output Consistency, but less likely to do so (with higher threshold estimates) on Coherence and Relevance. The estimated factor correlations below suggested high correlations between dimensions, especially for coherence and relevance:

|  | Coherence | Consistency | Fluency |
|---|---|---|---|
| Consistency | .51 | – | – |
| Fluency | .56 | .68 | – |
| Relevance | .86 | .64 | .53 |

## A.5 Multitrait-Multimethod Table for G-EVAL and expert-based metrics

Table 2 presents the multitrait-multimethod table on the four dimensions, Coherence, Consistency, Fluency, and Relevance, for three separate rating methods: expert-based ratings (average factor score across three raters), G-EVAL-3.5, and G-EVAL-4.

## A.6 Residual Analysis

We further performed principal component analysis on the residuals of the automatic evaluations,

which capture the unexplained variance by the 4 dimensions' factor scores. Plot of the first two principal components is shown in Fig. 7. Here, visual clusters of evaluation metrics are found, suggesting that select metrics likely tapped on common additional dimensions. The unexplained residual variance may guide future investigation on discovering other quality signals in summarization tasks.

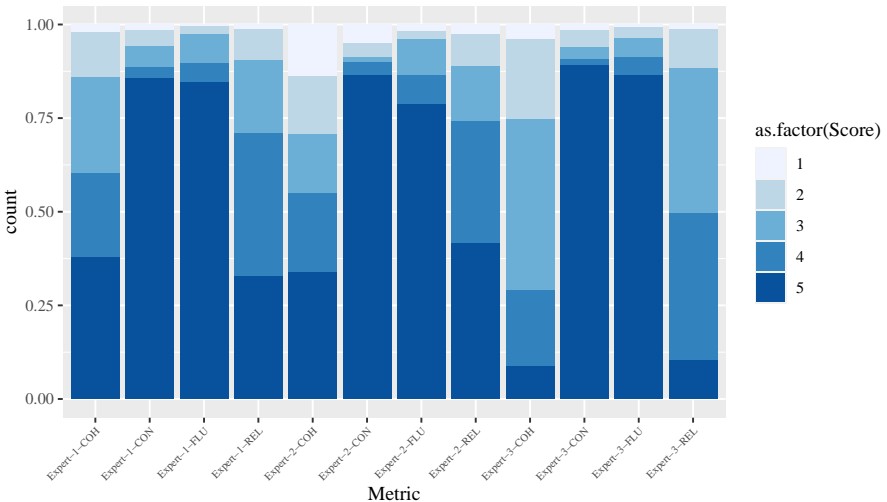

Figure 6: Distribution of 5-point scale expert ratings.

| | | Expert | | | | G-EVAL-3.5 | | | | G-EVAL-4 | | | |
|---|---|---|---|---|---|---|---|---|---|---|---|---|---|
| | | Coherence | Consistency | Fluency | Relevance | Coherence | Consistency | Fluency | Relevance | Coherence | Consistency | Fluency | Relevance |
| Expert | Coherence | .98 | *0.32* | *0.46* | *0.69* | **0.50** | 0.32 | 0.49 | 0.53 | **0.65** | 0.46 | 0.65 | 0.60 |
| | Consistency | | .97 | *0.57* | *0.46* | 0.59 | **0.56** | 0.54 | 0.59 | 0.62 | **_0.72_** | 0.50 | 0.63 |
| | Fluency | | | .97 | *0.59* | 0.63 | 0.66 | **_0.71_** | _0.75_ | _0.75_ | 0.47 | **0.60** | _0.71_ |
| | Relevance | | | | .97 | 0.60 | 0.46 | 0.65 | **0.63** | _0.75_ | 0.65 | _0.72_ | **_0.76_** |
| G-EVAL-3.5 | Coherence | | | | | 0.96 | *0.59* | *_0.87_* | *_0.79_* | **0.65** | 0.54 | 0.53 | 0.63 |
| | Consistency | | | | | | 0.90 | *0.60* | *_0.71_* | 0.56 | **0.51** | 0.35 | 0.54 |
| | Fluency | | | | | | | 0.95 | *_0.81_* | 0.66 | 0.47 | **0.51** | 0.65 |
| | Relevance | | | | | | | | 0.95 | _0.74_ | 0.57 | 0.53 | **0.69** |
| G-EVAL-4 | Coherence | | | | | | | | | 0.98 | *0.69* | *_0.71_* | *_0.90_* |
| | Consistency | | | | | | | | | | 0.96 | *0.54* | *_0.71_* |
| | Fluency | | | | | | | | | | | 0.97 | *0.69* |
| | Relevance | | | | | | | | | | | | 0.96 |

Table 2: Multitrait-Multimethod table of the pairwise correlations between expert rating factor scores, GPT3.5-based scores, and GPT4-based scores.
*Notes:* Entries in bold are the correlations of ratings on the same dimension by different methods. Entries in italic are the correlations of the ratings on different dimensions using the same method. Except for reliability coefficients, entries over .7 are underscored. While the cutoff is arbitrary, underscoring is more desirable for bolded entries (indicating good convergent validity) and less so for italic entries (indicating method bias, worse divergent validity).

| | Loading | | | | Threshold | | | |
|---|---|---|---|---|---|---|---|---|
| | coherence | consistency | fluency | relevance | 2 | 3 | 4 | 5 |
| expert_1_coherence | 0.90 | - | - | - | -2.03 | -1.08 | -0.26 | 0.31 |
| expert_2_coherence | 0.88 | - | - | - | -1.09 | -0.55 | -0.12 | 0.42 |
| expert_3_coherence | 0.76 | - | - | - | -1.74 | -0.67 | 0.56 | 1.37 |
| expert_1_consistency | - | 0.97 | - | - | -2.15 | -1.56 | -1.20 | -1.07 |
| expert_2_consistency | - | 0.98 | - | - | -1.63 | -1.36 | -1.27 | -1.10 |
| expert_3_consistency | - | 1.00 | - | - | -2.16 | -1.55 | -1.33 | -1.23 |
| expert_1_fluency | - | - | 0.98 | - | -2.60 | -1.93 | -1.26 | -1.02 |
| expert_2_fluency | - | - | 0.88 | - | -2.12 | -1.74 | -1.09 | -0.80 |
| expert_3_fluency | - | - | 0.92 | - | -2.49 | -1.81 | -1.35 | -1.10 |
| expert_1_relevance | - | - | - | 0.79 | -2.25 | -1.30 | -0.55 | 0.45 |
| expert_2_relevance | - | - | - | 0.85 | -1.95 | -1.23 | -0.65 | 0.22 |
| expert_3_relevance | - | - | - | 0.64 | -2.28 | -1.19 | 0.01 | 1.26 |

Table 3: Factor loading and score category threshold estimates for the 4-factor confirmatory model of ordinal expert ratings.

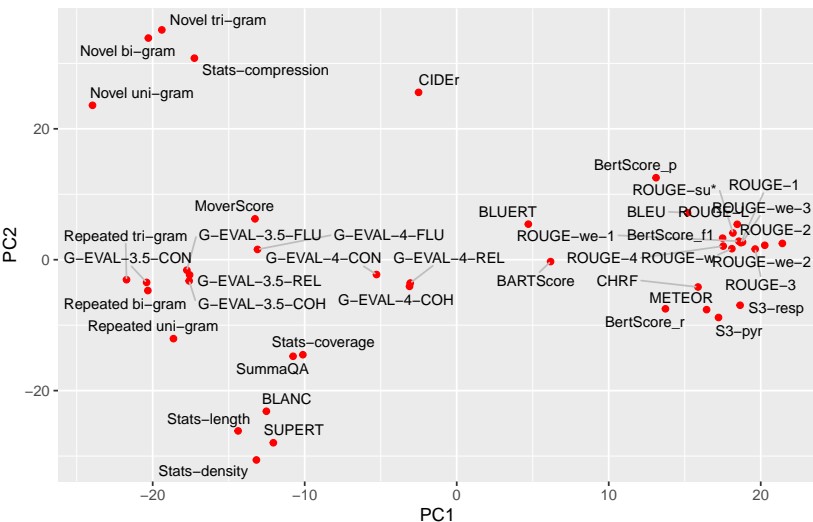

Figure 7: Plot of the first two principal components of the residuals of the 4-factor model.

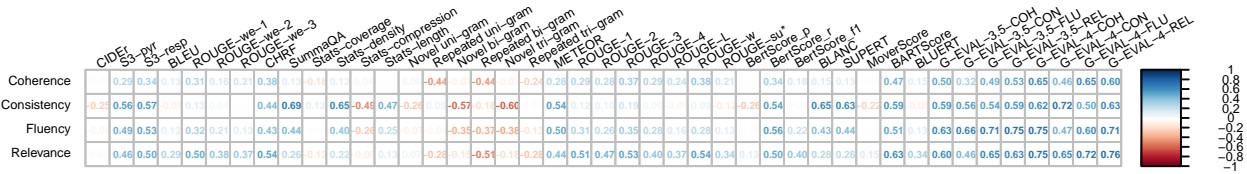

Figure 8: Concurrent validity coefficients of the metric-based scores in predicting the four expert-rated dimensions' factor scores. Values are based on Kendall's $\tau$.