# OpenReview forum: "Evaluating Evaluation Metrics: A Framework for Analyzing NLG Evaluation Metrics using Measurement Theory"
_EMNLP/2023/Conference — EMNLP 2023 Main_

### Official Review · Reviewer_DnGL · 2023-08-01

**Soundness:** 4

**Excitement:**

4: Strong: This paper deepens the understanding of some phenomenon or lowers the barriers to an existing research direction.

**Missing References:**

- Sec 3.1: https://aclanthology.org/2022.findings-emnlp.108.pdf developed a theoretical framework for measuring the unreliability of a metric, as you mention them here

**Paper Topic And Main Contributions:**

This paper shows how to apply measurement theory to evaluate the quality of metrics for text generation. They showcase their approach for text summarization with 16 sota metrics on the SummEval dataset.

**Reasons To Accept:**

+ nice combination of existing methods (from measurement theory) to a very relevant current topic (evaluation of metrics for NLG)
+ nice and intuitive showcase

**Reasons To Reject:**

none

**Reproducibility:**

5: Could easily reproduce the results.

**Reviewer Confidence:**

4: Quite sure. I tried to check the important points carefully. It's unlikely, though conceivable, that I missed something that should affect my ratings.

**Typos Grammar Style And Presentation Improvements:**

- l.24: term "NLG" is traditionally used for data2text tasks, maybe mention this in a footnote
- l.42: whitespace missing after "quality"
- l.71: High correlation with human judgements does not always imply a good metric, as has been shown in https://aclanthology.org/2022.acl-short.85.pdf.
- l.130: other -> on
- l.298: there is no T2 in the equation above
- l.305ff: unclear formulation: how would asses that a metric "deviates for a SNGLE datapoint"? Lateron, you use different subsets, which makes more sense
- l.320: the notation is confusing: you say that Y_j is "the observed score (of each model) on the jth data point", which is a set of scores. How can you sum over this set lateron?
- Sec. 3: I think that using lower case letters (n, j) for numbers and upper case letter for scores (X, Y) would improve readibility a lot
- l. 315ff: i do not see how the subsequent computation of alpha correlates to your idea to use the differences on different subsets of test data, as mentioned here
- l.342: whitespace missing
- l.433: "introduce" sounds like you invented these MTMM, but lateron you reference a paper from 1959.
- l. 487: maybe use a ' in the brackets for better readibility: \lambda_{jk}'s
- l. 635: discriminative -> discriminate
- Section References: use proper upper-casing, e.g. ACL in second reference, NLG in Howcroft et al etc.
- add a reference to good introduction to measurement theory (maybe Allen et al, or something similar)

---

> ### Author Rebuttal · Authors · 2023-08-29
>
> Thanks for your encouraging review! We have extended our related work section to incorporate your suggested paper. Here, we want to clarify a few questions you raised.
>
> **l.305ff: unclear formulation: how would assess that a metric "deviates for a SINGLE datapoint"? Lateron, you use different subsets, which makes more sense**
>
> Thanks for raising this question. To clarify this, consider the following example: Suppose different models are compared on their average eval score across 100 news summarization tasks, and across models, the average score standard deviation is 0.1. Further suppose that for a specific model, the score on each specific news article deviates from the average by 0.4. In this case, the metric internal consistency is low — For a specific model, the average score on any subset of the 100 tasks is expected to be sensitive to the choice of included examples. A good proportion of difference across the two evaluated models’ average scores would also be attributed to noise, i.e., random fluctuations in scores across news articles. We will make it clear in the next version.
>
> **l.320: the notation is confusing: you say that Y_j is "the observed score (of each model) on the jth data point", which is a set of scores. How can you sum over this set lateron?**
>
> Thanks for the question. Y_j is the eval score on a single data point, e.g., a single news article. Therefore, for a test set consisting of 100 articles, j ranges from 1 to 100, and a model’s overall score on the metric is the average (or similarly, sum) across the 100 Y_js. We will clarify it in the revision.
>
> **l. 315ff: i do not see how the subsequent computation of alpha correlates to your idea to use the differences on different subsets of test data, as mentioned here**
>
> We briefly explain the intuition behind alpha: By the property of variance of sums (i.e., Bienaymé's identity), the numerator of Equation (2) is larger when the pairwise covariance between scores on each single data point (e.g., news article) is larger. In other words, alpha is large when models’ scores on each pair of data points are highly correlated. More specifically, consider evaluation of summarization performance and 100 articles in the test set: If on any two news articles, different candidate models’ rankings on an eval metric are the same, then the pairwise single data point score correlation is expected to be high, and alpha will be high. On the other hand, if the ranking of different models’ eval scores differs remarkably across articles, the pairwise correlation of eval scores on single data points would be low, and thus alpha is low.
> \
> \
> \
> We also fixed the grammatical errors and format issues. Thanks again for your receptive review.

---

### Official Review · Reviewer_rppc · 2023-08-02

**Soundness:** 4

**Excitement:**

3: Ambivalent: It has merits (e.g., it reports state-of-the-art results, the idea is nice), but there are key weaknesses (e.g., it describes incremental work), and it can significantly benefit from another round of revision. However, I won't object to accepting it if my co-reviewers champion it.

**Missing References:**

The related work section is quite succinct. It is missing a more thorough discussion on previous work that aimed to evaluate evaluation metrics. For instance:

- Caglayan et al. (2020). Curious case of language generation evaluation metrics: A cautionary tale.
- Hanna & Bojar (2021). A fine-grained analysis of BERTScore.
- Sai et al. (2021). Perturbation CheckLists for evaluating NLG evaluation metrics.
- Sai et al. (2022). A survey of evaluation metrics used for nlg systems.

More importantly, the paper ignores important previous literature where other researchers have also aimed to look beyond correlation scores:

- Ni'mah et al. (2023). NLG Evaluation Metrics Beyond Correlation Analysis: An Empirical Metric Preference Checklist.
- Fomicheva & Specia (2019). Taking MT Evaluation Metrics to Extremes: Beyond Correlation with Human Judgments.

**Paper Topic And Main Contributions:**

This paper describes the application of measurement theory, originating from psychology and educational sciences, to assess the quality of commonly used NLG metrics. These metrics help to more systematically analyze and evaluate NLG metrics, and compare them to a human evaluation baseline. These tests are also more extensive than just correlation, which has been used previously for this task. The authors focus on the summarization task (CNN/Daily Mail benchmark dataset) for their assessment of various metrics (reference-based as well as reference-free).

**Questions For The Authors:**

Overall, I think the authors did thorough work in translating developments in measurement theory to be applicable to NLG. However, I did run into a few questions and uncertainties regarding the reporting and methodological choices. I will list these below.

- In general (mostly §2): I think the argument for why measurement theory should be applied in this context is not sufficiently argued. For instance, it is not clear to me what exactly the benefits are of applying measurement theory in this context, and what makes the application of measurement theory in this context so unique.
- In general: Most of the described measurement theory tests have been developed for human evaluation. To what degree do they also apply to metrics? For instance, stability seems irrelevant for most metrics, as the formula to calculate the score is fixed for these metrics.
- In general (mostly §4): Why are the authors looking specifically at evaluating summarization metrics and the CNN/Daily Mail dataset? What makes this task/dataset relevant? And how generalizable are the results of this study for other tasks/datasets?
- In general (mostly §4): While the authors go into detail about the results per measurement criteria, I am missing a higher-level conclusion from the case study's results: which metrics perform well, and which do not perform well? Why do they (not) perform well? What kind of information did we obtain that we would not have obtained if we would have just looked at correlations? I think such a higher level conclusion of their case study would greatly help other researchers to understand their framework and make it more attractive to use said framework.
- 551-553: "for less stable LLM-based metrics, the evaluator should consider applying the metrics multiple times" -->  Is it really less stable as the lowest it scores are around 0.98?
- 582-584: "Inter-rater agreements on the same dimension were high (.48 − .89) in general but lower for Relevance (.48 − .58)." --> Which inter-rater agreement metric was used here?
- 610-618: "Since Coherence and Relevance are distinct constructs by definition (Fabbri et al., 2021), the conflated validity structure indicates potential issues in the expert rating process. One source of such systematic measurement error may come from unclear instructions or the expert rater’s individual leniency. Such issues may cascade if new automatic evaluation metrics were trained or validated on this dataset." --> Is there a way to fix this, post-hoc? For instance, by merging these criteria into one single construct?

**Reasons To Accept:**

- Interesting framework that addresses an important task: estimating the quality of commonly used metrics.
- Thorough description of measurement theory and how it is applied, enabling other researchers to use this framework for more cases.
- Case study extensive in regards to metrics (a good selection of reference-free and reference-based metrics).

**Reasons To Reject:**

- Uncertainty to what degree measurement theory (which has been developed to assess human evaluation) can also be applied to automatic metrics.
- Case study somewhat limited in scope in regards to the task (only summarization).
- Limited relevant work section.

**Reproducibility:**

4: Could mostly reproduce the results, but there may be some variation because of sample variance or minor variations in their interpretation of the protocol or method.

**Reviewer Confidence:**

4: Quite sure. I tried to check the important points carefully. It's unlikely, though conceivable, that I missed something that should affect my ratings.

**Typos Grammar Style And Presentation Improvements:**

- 042: "language quality(Nema and Khapra" --> Missing a space before the opening parenthesis
- 046-051: "(e.g., ROUGE (Lin, 2004), BLEU (Papineni et al., 2002)) to embedding-based metrics (e.g., BERTScore (Zhang et al., 2019), MoverScore (Zhao et al., 2019)) and to end-to-end metrics (e.g., BLEURT (Sellam et al., 2020), G-Eval (Liu et al., 2023))" --> This is a personal opinion to some degree, but I would try to avoid nested parentheses.
- 102: "systemically" --> systematically
- 131: "other the other hand" --> on the other hand
- 184: "Evalaution" --> Evaluation
- 519: "on CNN/Daily Mail benchmark dataset" --> on the CNN/Daily Mail benchmark dataset

---

> ### Author Rebuttal · Authors · 2023-08-29
>
> Thank you for your very constructive feedback!  We will address your comments and will acknowledge the limitations of the paper in our revisions.
>
>
> We hope our clarifications will address your concerns.
>
>
> **Uncertainty to what degree measurement theory (which has been developed to assess human evaluation) can also be applied to automatic metrics.**
>
> In the revision, we will elaborate on the rationale behind extending measurement theory to the NLG evaluation context, in a few aspects:
> - *Similar measurement objective:* Both the measurement of human capability and NLG model’s capability use observable tasks to estimate unobservable capability. The need for measuring NLG’s model unobservable capability is increased as more models are referred to as general-purpose model. By bridging measurement theory for human capabilities to NLG evaluation, our work provides a more principled way of reasoning the metrics design, analyzing metric’s behavior, and improving evaluation methods.
> - *Similar assumptions, tasks, and goals:* Both the measurement of human and the evaluation of NLG evaluate output on each task with a rubric (eval metric) defined based on assumptions of what constitutes successful task completion, compute an aggregate score (e.g., sum/average) on a set of specific tasks, and use the aggregate score for inferences about candidate's (model’s) capability in specific domains and future performance on tasks beyond what’s included in the test (or test set).
> - *Theory driven:* Measurement theory provides a theory-driven framework for enumerating (1) the assumptions made in developing, using, and interpreting (both automatic and human-based) NLG evaluation metrics, and (2) the different sources of measurement error (e.g., task-, rater-, and metric-specific) that lead to random fluctuation in a model’s score on a metric. It also offers a methodology that guides collecting and analyzing NLG evaluation data, to inform users on (1) uses and interpretations of eval scores supported by empirical evidence, and (2) to what extent differences across models on an eval can be attributed to signal versus noise.
>
> We also want to clarify that the measurement theory applies to both human evaluation and automatic scores. The measurement theory does not make assumptions on who derived the score or how the score is calculated. In our framework, an evaluation metric is a scoring function that transforms a model output into a numerical score. The process to derive the score could be human evaluation, rule-based metrics, learned metrics, or LLM-based metrics.
>
>
>
>
> **Stability seems irrelevant for most metrics, as the formula to calculate the score is fixed for these metrics.**
>
>
> As we see increasing use of automatic evaluation metrics with built-in stochasticity (e.g., LLM-based metrics), the stability of an eval in producing consistent scores on an output from one replication to another will be increasingly relevant. We will add this point in the revision.
>
>
> **Case study somewhat limited in scope in regard to the task (only summarization). Why are the authors looking specifically at evaluating summarization metrics and the CNN/Daily Mail dataset? What makes this task/dataset relevant? And how generalizable are the results of this study for other tasks/datasets?**
>
>
> The case study is limited in this scope. However, the goal of this case study is to offer a demonstration of how to apply our proposed framework and show the framework’s practical utility. As noted in the paper, our evaluation of metrics should be interpreted as dependent on the benchmark design, and is not meant to be generalizable (though some generalizations can be derived by studying multiple benchmarks, which we plan to explore in future work). We will further acknowledge the limited scope of this case study, and provide the following rationales for choosing this task as the case study.
> - *Importance and challenge of summarization tasks:* Summarization is a critical and complex task in the field of NLP, requiring a model to process the content, grasp the context, extract salient information, and output in a fluent manner. Given the complex process, a model’s summarization capability cannot be directly observed, which aligns with the assumption of measurement theory.
> - *Evaluation difficulties:* Assessing the quality of summaries is inherently challenging, our community has invested a lot of effort in evaluating summarization evaluation metrics and developing novel metrics. Although our framework could be applied to other NLG metrics, we hope our case study could contribute to the summarization tasks and aid the ongoing effort.
> - *Real-world relevance:* Summarization is a common task in various applications, including news summarization, meeting summarization, and long documents (e.g., academic papers) summary. Selecting informative evaluation metrics is important to inform model development and model selection. Therefore, we want to ground our case study in this important field.
> - *Dataset significance:* The SummEval dataset, which includes valuable expert ratings, has been widely used to train and validate novel metrics in summarization. Although it is only built on CNN/DM, it is a common dataset used in model development. We are also aware of some recent novel evaluation datasets, such as Seahorse [1]. We did plan to expand our analysis of those datasets as part of our future work.
>
> To also encourage others to apply our framework to other benchmarks to inform generalizability, we will release a  Python toolkit based on our framework to encourage the community to use our framework to evaluate evaluation metrics for different datasets and tasks. We are currently working on building a centralized benchmark for analyzing evaluation metrics under multiple datasets and multiple tasks. We hope such a benchmark will allow us to gain insights into what context a metric is a good metric and allow practitioners to select metrics for their own use cases.
>
>
> **Limited relevant work section.**
>
> We appreciate your comments and suggestions. We have extended our related work with the extra page to provide a more comprehensive view of how our framework is situated in prior work and make our contribution more clear.
>
>
> **High-level conclusion from the case study results: which metrics perform well, and which do not perform well? Why do they (not) perform well? What kind of information did we obtain that we would not have obtained if we would have just looked at correlations?**
>
>
> We will add another paragraph at the end of the case study to summarize our findings. Here, we gave a brief summary of the major takeaway,
>
>
> - *Metric Stability:* Compared to other metrics,  LLM-based metrics show lower stability. Also, by convention (.9), most of the LLM-based metrics can be considered stable. Some LLM-based metrics such as G-EVAL-3.5-Consistency exhibit unsatisfactory stability.
>
>
> - *Metric Consistency:* For the ROUGE family, a longer n-gram makes the metric less reliable and more prone to potential data perturbations in the test dataset. Both metric stability and consistency coefficients could help us compare true model differences by estimating the standard error on the metric score.
>
>
> - *Metric Construct Validity:* Through MTMM and factor analysis, we found in SummEval, expert ratings on Coherence and Relevance are conflated. The conflated validity construct may cascade to metrics that are trained or validated on this dataset.
>
>
> - *Metric Concurrent Validity:* We found although BARTScore and G-Eval are sensitive to detecting
> quality signals in all four expert-rated dimensions, the lack of variance in the validity coefficients indicates their lack of capability to discriminative different dimensions.
>
>
>
>
>
>
>
>
> **551-553: "for less stable LLM-based metrics, the evaluator should consider applying the metrics multiple times" --> Is it really less stable as the lowest it scores are around 0.98?**
>
>
> In the appendix, we report stability estimates of all metrics. It can be seen that for some specific GEVAL dimensions other than the average across dimensions,  the metric stability is below .9 for GEVAL-3.5-Consistency. In addition, we want to clarify the attenuation effect of in-stable metrics. We believe although current metrics achieve high stability, metric stability is an important aspect of a metric as an in-stable metric may amplify its measurement error from other aspects of the metric.
>
>
> **582-584: "Inter-rater agreements on the same dimension were high (.48 − .89) in general but lower for Relevance (.48 − .58)." --> Which inter-rater agreement metric was used here?**
>
>
>  We calculated Kendall's rank correlation in the model-level average scores. We will clarify it in our paper.
>
>
> **610-618: Is there a way to fix the conflated validity structure, post-hoc? For instance, by merging these criteria into one single construct?**
>
>
> When the conflated validity structure occurs, it often indicates the evaluator (i.e., expert rater) can not differentiate the two constructs very well. In this case, a few diagnostic or remedial steps should be taken:
> - Revisit the definition of those dimensions in terms of whether they are conceptually distinct. If they are not, consider revising the theoretical framework.
> - Revisit the instructions provided for human-annotation.
> - Revisit the test set to examine whether the specific tasks employed for evaluating the model are capable of distinguishing model capability on the two dimensions.
>
> If the issue can not be fixed, we would suggest the community rethink the conceptualization of each individual aspect of summarization quality. For example, in a few recent works, such as [1,2], the researcher suggests new constructs for summarization. Post-hoc-ly speaking, merging two constructs is one possibility, if we conclude that two factors are explained by the same underlying construct. However, we should also rule out potential alternative explanations, in particular, problematic rubrics (e.g., ambiguous directions provided to expert raters) or problematic test set that cannot distinguish model capabilities on separate dimensions.
> \
> \
> \
> We want to thank you again for your valuable and detailed comments and for being receptive to our idea. We will also do another thorough pass to fix all minor issues. We hope our rebuttal addressed your comments.
>
> [1]Clark, E., Rijhwani, S., Gehrmann, S., Maynez, J., Aharoni, R., Nikolaev, V., ... & Parikh, A. P. (2023). SEAHORSE: A Multilingual, Multifaceted Dataset for Summarization Evaluation. arXiv preprint arXiv:2305.13194.
>
> [2]Liu, Y., Fabbri, A. R., Liu, P., Zhao, Y., Nan, L., Han, R., ... & Radev, D. (2022). Revisiting the gold standard: Grounding summarization evaluation with robust human evaluation. arXiv preprint arXiv:2212.07981.

---

### Official Review · Reviewer_QqGJ · 2023-08-16

**Typos Grammar Style And Presentation Improvements:** 1. Line 041
**Soundness:** 4

**Excitement:**

4: Strong: This paper deepens the understanding of some phenomenon or lowers the barriers to an existing research direction.

**Missing References:**

Daniel Deutsch, Rotem Dror, and Dan Roth. 2021. A Statistical Analysis of Summarization Evaluation Metrics Using Resampling Methods. Transactions of the Association for Computational Linguistics, 9:1132–1146.

Manik Bhandari, Pranav Narayan Gour, Atabak Ashfaq, Pengfei Liu, and Graham Neubig. 2020. Re-evaluating Evaluation in Text Summarization. In Proceedings of the 2020 Conference on Empirical Methods in Natural Language Processing (EMNLP), pages 9347–9359, Online. Association for Computational Linguistics.


**Paper Topic And Main Contributions:**

This paper focuses on evaluation metrics in natural language generation, proposing a new framework for evaluating these metrics inspired by measurement theory. The paper introduces to NLG desiderata and methods from educational and psychological testing and proposes a framework for analyzing these metrics. It also shows how these evaluation approaches may be used in practice by performing a case study on the SummEval dataset and comparing the performance of different metrics on it.

**Questions For The Authors:**

A. Since being sensitive to signals in the four SummEval dimensions alone doesn’t make for a good metric due to lack of discrimination between the dimensions, have you tried creating a weighted combination of the two factors to evaluate the overall quality of a metric?

B. It would be helpful to rank all metrics by their overall quality in order to understand which type generally tends to work better (lexical vs. semantic etc.) and if there is a certain direction in which metric creation research should focus.

C. Why are the self-correlations of values in Table 1 not equal to 1.0? E.g. for Expert 1 Coherence the value is 0.96.

D. Did you perform any case study on a different kind of task (e.g. more creative text generation) to compare how the metrics' behavior varies between them?

**Reasons To Accept:**

As NLG models improve over time, finding good metrics to accurately measure the quality of their output in a way that best approximates their real-world performance is an important task. This paper presents an interesting idea from educational testing that breaks down the required qualities of a good metric and shows how they relate to different factors. The paper is well written and the ideas are communicated clearly. The authors also present a case study using text summarization as an example task and show how different metrics score along different dimensions, and what this means for their robustness and validity. The authors plan to make the code for their framework publicly available as well.

**Reasons To Reject:**

There can be other sources of error beyond those discussed in the paper (and an investigation of where they occur in the evaluation process), although the authors concede this in the limitations section already. The related work section could be more comprehensive and discuss the merits and limitations of prior works in more detail.

**Reproducibility:**

4: Could mostly reproduce the results, but there may be some variation because of sample variance or minor variations in their interpretation of the protocol or method.

**Reviewer Confidence:**

3: Pretty sure, but there's a chance I missed something. Although I have a good feel for this area in general, I did not carefully check the paper's details, e.g., the math, experimental design, or novelty.

---

> ### Author Rebuttal · Authors · 2023-08-29
>
> Thank you for your valuable feedback. We agree with your comments and we are revising our manuscript to reflect those changes accordingly.
>
> Please find hereafter our answers to your questions:
>
> **Other sources of error beyond those discussed in the paper and an investigation of where they occur in the evaluation process.**
>
> We fully agree that there are other sources of measurement issues, including the choice of dataset, instructions for human annotators, score aggregation methods, etc. As we mentioned in the paper, the validity and reliability of an evaluation process depend on every individual evaluation component and related design choices. The primary goal of our paper is to bridge measurement theory to model evaluation and focus on two important sources of measurement errors, i.e., validity and reliability, that have been extensively studied in measurement theory but less so for model evaluation. Our current framework focuses on evaluation metrics, given the importance of metrics and their downstream implications on model comparison. In our future work, we plan to study measurement theory in the broader context of model evaluation, identify measurement issues beyond metrics, and propose new evaluation methods. In the revised version, we will further acknowledge the scope of the current framework and highlight other potential sources of measurement errors.
>
> **The related work section could be more comprehensive and discuss the merits and limitations of prior works in more detail.**
>
> We want to thank you for suggesting other related work. In the revised version, with the extra page, we will discuss the prior work and its connection with our current work in more detail.
>
> **Since being sensitive to signals in the four SummEval dimensions alone doesn’t make for a good metric due to lack of discrimination between the dimensions, have you tried creating a weighted combination of the two factors to evaluate the overall quality of a metric?**
>
> We did not try to combine those two factors as a single quality dimension in this case study. Although a combined score may be more straightforward to compare metrics, it is less informative in diagnosis. For example, for the discrimination power, troubleshooting often needs to know where the undesirable correlation comes from. Also, the weighting function for discrimination power on individual dimension and general sensitivity is often context dependent. We need future studies to understand how to formulate such a weighing function.
>
> **It would be helpful to rank all metrics by their overall quality in order to understand which type generally tends to work better (lexical vs. semantic etc.) and if there is a certain direction in which metric creation research should focus.**
>
> In the current case study, we focused on demonstrating how our framework can be applied to evaluate different metrics rather than providing an overall ranking. In addition, we decided not to rank different metrics based on the single case study. Given the performance of a metric is dependent on the chosen dataset and task (as we highlighted in our paper), ranking metrics based on this study may generate misleading interpretations (e.g., metric’s general performance). However, we agree that a benchmark for metrics is based on the analysis of multiple datasets and multiple tasks will provide more informative comparisons. Such a benchmark will allow researchers to answer questions like in what context a metric is a good metric and practitioners to select metrics for their own use cases. This is one of our ongoing work and we hope to share the results soon.
>
> **Why are the self-correlations of values in Table 1 not equal to 1.0? E.g., for Expert 1 Coherence the value is 0.96.**
>
> The diagonals of the MTMM table present the metric internal consistency estimates of each evaluation metric based on Equation (2), which are bounded above by 1. This informs the evaluation of the extent to which the observed score correlations on a pair of variables are attenuated by the reliability of each variable.  We will add this to the caption for clarity.
>
> **Did you perform any case study on a different kind of task (e.g. more creative text generation) to compare how the metrics' behavior varies between them?**
>
> We have not applied our framework to another case study. However, we agree with the importance of analyzing metrics on different kinds of tasks and the comparison will better help us answer the question of in what context an evaluation metric is good. Towards this effort, we will release a Python toolkit based on our framework to encourage a community effort to deepen our understanding of how evaluation metrics perform in different tasks.
> \
> \
> \
> We hope that our responses have addressed your concerns and questions. We have already updated our draft with a more comprehensive related work and fixed all grammatical and formatting issues you identified. Thanks again for your feedback.

---

### Meta-Review · Area_Chair_zE7C · 2023-09-13

**Recommendation:** 5

**Metareview:**

This paper focuses on evaluation metrics in natural language generation, proposing a new framework for evaluating these metrics inspired by measurement theory. The authors have devised a new framework, influenced by measurement theory, to evaluate these metrics. The ideas are effectively communicated, and the paper is well-written. Additionally, the authors have presented a case study on text summarization to demonstrate the scoring of different metrics on various dimensions and their robustness and validity implications. The paper's only suggested improvements are to expand the related work section and add the missing references mentioned by the reviewers, which can be easily addressed with an additional page. All in all, I regard the paper as of high quality and a significant contribution to its field.

---

### Decision · Program_Chairs · 2023-10-07

**Decision:**

Accept-Main

**Comment:**

This paper focuses on evaluation metrics in natural language generation, proposing a new framework for evaluating these metrics inspired by measurement theory. The authors have devised a new framework, influenced by measurement theory, to evaluate these metrics. The ideas are effectively communicated, and the paper is well-written. Additionally, the authors have presented a case study on text summarization to demonstrate the scoring of different metrics on various dimensions and their robustness and validity implications. The paper's only suggested improvements are to expand the related work section and add the missing references mentioned by the reviewers, which can be easily addressed with an additional page. All in all, I regard the paper as of high quality and a significant contribution to its field.